# On the Weldability of Thick P355NL1 Pressure Vessel Steel Plates Using Laser Welding

**DOI:** 10.3390/ma14010131

**Published:** 2020-12-30

**Authors:** Jiří Čapek, Karel Trojan, Jan Kec, Ivo Černý, Nikolaj Ganev, Stanislav Němeček

**Affiliations:** 1Department of Solid State Engineering, Faculty of Nuclear Sciences and Physical Engineering, Czech Technical University in Prague, Trojanova 13, 120 00 Prague, Czech Republic; karel.trojan@fjfi.cvut.cz (K.T.); nikolaj.ganev@fjfi.cvut.cz (N.G.); 2Laboratory of Material Properties, SVÚM a.s., Tovární 2053, 250 88 Čelákovice, Czech Republic; kec@svum.cz (J.K.); ivo.cerny@seznam.cz (I.Č.); 3Department of Laser Material Processing, RAPTECH s.r.o., U Vodárny 473, 330 08 Zruč-Senec, Czech Republic; nemecek@raptech.cz

**Keywords:** laser welding, pressure vessel steel, microstructure, X-ray and neutron diffraction, high-cycle fatigue tests

## Abstract

Pipeline transport uses millions of kilometers of pipes worldwide to transport liquid or gas over long distances to the point of consumption. High demands are placed, especially on the transport of hazardous substances under high pressure (gas, oil, etc.). Mostly seamless steel pipes of various diameters are used, but their production is expensive. The use of laser-welded pipes could significantly reduce the cost of building new pipelines. However, sufficient mechanical properties need to be ensured for welded pipes to meet stringent requirements. Therefore, laser-welded 10 mm thick pressure vessel steel plates were subjected to various mechanical tests, including high-cycle fatigue tests. Furthermore, the microstructural parameters and the state of residual stresses were determined using X-ray and neutron diffraction, which could affect fatigue life, too. The critical areas for possible crack initialization, especially in and near the heat-affected zone, were found using different tests. The presented results outline the promising application potential of laser welding for the production of pipes for high-pressure pipelines.

## 1. Introduction

Welding has been widely used in fabrication industries producing ships, trains, steel bridges, pressure vessels, and more since the First World War. Increasing demands are being placed on the mechanical properties and durability of welds that are used to connect two or more components. There is also an increasing demand for productivity and cost-effectiveness of welding, which leads to the use of modern and progressive methods, including laser welding, which has an advantage at high welding speeds, a low thermal load of the surrounding material, precision and strength of the weld, and the possibility of joining components with a wide range of thickness (0.01 to 50 mm) [1,2].

Pipeline transport uses pipes to transport liquid or gas over long distances to the point of consumption. Millions of kilometers of pipes are used around the world for various applications. High demands are placed, especially on the transport of hazardous substances under high pressure (gas, oil, etc.). Therefore, the use of welded pipes could significantly reduce the cost of building new pipelines, if they replaced seamless pipes, which are expensive to manufacture. However, sufficient mechanical properties need to be ensured for welded pipes to meet stringent requirements to prevent accidents and natural disasters. Laser welding, with its advantages described above, could find an application in the welding of longitudinal welds on pipes in production. Another advantage of laser welding in comparison to conventional welding methods is the robotic automation of the welding process and to a defined wall thickness without additional material. Compared to electron welding, which also has excellent properties, there is no need to use a vacuum chamber, which would have to be up to 30 m long. However, welding of the segments during the construction of the pipeline itself will continue to be carried out mainly by manual welding, as a sufficiently precise fit cannot be guaranteed in the field.

With the application of new welding technology, it is necessary to describe the influence on the basic mechanical parameters. Tensile, Charpy impact, and hardness tests are usually an integral part of the set of performed tests. Furthermore, fatigue loading is a very important factor affecting service safety and reliability, mainly of dynamically loaded structures. Therefore, an evaluation of fatigue resistance together with the evaluation of damage mechanisms and affecting factors has to be considered in such cases. Note that an investigation of fatigue properties and phenomena is still missing in some works aimed at laser welding, e.g., high-pressure pipelines [3].

It is necessary to mention that the final mechanical properties are strongly dependent on the welding parameters, e.g., [4,5]. Zhang et al. [5] further found that some laser welding parameters could cause the mechanical properties of weld metal (WM) to not reach the properties of the base metal (BM). Mainly welding speed and laser power have a strong effect on the residual stresses (RS) among the studied parameters, e.g., increasing weld speed. The piping with laser power reduction decreases residual stresses with hardness and increases the toughness. However, it is important to monitor sufficient penetration and pressure fluctuations.

According to Guo et al. [6], the welding of low-alloy steel causes the generation of martensite in the WM and heat-affected zone (HAZ), which leads to an increase in hardness, but also a reduction of impact toughness in comparison with the BM. Moreover, laser welding causes microstructural changes of plastically deformed crystalline materials and decreases ductility and corrosion resistance due to the grain coarsening, carbide precipitation, and martensite formation [7]. These so-called microstructural notches are the critical areas for the potential crack initialization, e.g., at the interface between the WM and the HAZ.

Due to a heterogeneous application of energy and localized fusion that occurs during the welding process, high undesirable RS could be present in the region near the weld and in the weld itself. These RS are generated because of the superposition of thermal transformation processes [8] and could reach high values and subsequently cause fatigue or, in combination with crack-like defects, promote brittle fractures [9].

The influence of welding techniques on RS and the microstructural parameter FWHM (full width at half maximum) was investigated in Čapek et al. [10]. Comparable results for laser and electron welding were found; moreover, they were better than for conventional manual arc welding (MAG). The influence on the high-fatigue resistance was discussed, too, where a higher FWHM, i.e., higher microdeformation, higher dislocation density, or smaller crystallite size, could generally indicate a strengthening of the surface. The contribution Černý et al. [11] brought is the view of the high-cycle fatigue resistance of the laser welds. Partial attention was paid to the evaluation of crack initiation mechanisms and welding imperfections (internal lack of fusion, pores, etc.). These imperfections are a critical place for crack initialization, partly because of the reduction of the real effective cross-section. In the contribution of [12], the effect of re-calculation by considering only the real cross-section of welds without a lack of fusion or pores, as well as surface residual stresses, were discussed.

The research done by Moravec et al. [13] was focused on the evaluation of the laser welds under dynamic loading. It was shown that heat input has a place in the area of high-cycle fatigue. Higher heat input causes a longer thermal exposition; therefore, major changes could occur in the HAZ. The negative influence of these changes could have a very significant effect on the mechanical properties (hardness, toughness value, yield strength, etc.). Our team in [14] investigated the influence of a high-cycle fatigue test on the values of RS and microdeformation. The main conclusion was that the tensile RS could accelerate the fatigue initiation process and microdeformation could be a good indicator of future crack initialization and propagation. 

The purpose of this investigation was to analyze the relationship between microstructure, mechanical properties, and RS not only of WM but also of surrounding areas. This article describes a comprehensive evaluation of mechanical and fatigue properties of laser-welded pressure vessel steel plates. Specifically, the 10 mm thick double-sided square butt welds performed on P355NL1 steel plates for high-pressure vessels were analyzed. This fine-grained low-alloy carbon steel is widely used for the fabrication of high-pressure vessels, steam boiler parts, pressure piping, compressors, etc. Based on the results of this research, it would be possible to continue the development of laser welding technology for pipes, which would find applications in the construction of pressure pipelines.

## 2. Materials and Methods 

The P355NL1 hot-rolled steel plates of the dimensions 150 mm × 300 mm × 10 mm were studied. Flat plates were used for better processing of the results, especially fatigue tests. The chemical composition of this steel determined by glow-discharge optical emission spectroscopy is given in Table 1. The error at low concentrations was up to 10%. The determined chemical composition was in agreement with the European Standard EN 10028-3:2017 [15] with one exception, manganese, whose weight fraction was below the minimum value of 1.1 wt.% stated in the norm. This had a consequence in lowering the value of the carbon equivalent to the value of 0.21, according to IIW (International Institute of Welding).

The double-sided square butt weld (the top side was welded first) was performed by the Laserline LDM3000-60 (Laserline GmbH, Koblenz, Germany) from the company RAPTECH s.r.o. The most important welding parameters, such as laser power *P*, welding speed *v*, welding mode, and laser wavelength *λ*, are given in Table 2.

For metallographic analysis, the sample was first ground and afterwards polished. The steel structure was induced by etching in 2% Nital (98 mL Ethanol + 2 mL HNO_3_). The analysis was performed on the Zeiss Axio Observer light microscope (Carl Zeiss Microscopy GmbH, Berlin, Germany).

Lattice deformations of the diffraction lines *Kα_1_* of the planes *{211}* of the ferrite phase were analyzed using the Proto iXRD Combo diffractometer (Proto Manufacturing Inc., Taylor, MI, USA) in a ω-goniometer setup with chromium radiation. Diffraction angles 2*θ^hkl^* were determined by using the Gaussian function and Absolute peak method. The standard X-ray elastic constants *½s*_2_ = 5.75 TPa^−1^, *s*_1_ = −1.25 TPa^−1^, and the Winholtz and Cohen method [16] were used to calculate the surface RS. The FWHM parameter denoted the full width at half maximum of *Kα_1_* diffraction line.

The X’Pert PRO MPD diffractometer (Malvern Panalytical B.V., Almelo, The Netherlands) with cobalt radiation was used for the analyses by X-ray diffraction (XRD). The crystallite size (the size of coherently diffracted domains) and microdeformation were determined from the XRD patterns using the Rietveld refinement, performed in MStruct software(version from 2019) [17]. 

The sample was analyzed by XRD in both the perpendicular “T” and parallel “L” directions to the weld on both sides. The welded plate was analyzed in the middle area (see Figure 1). The irradiated volume was defined by experiment geometry, the effective penetration depth of the X-ray radiation (approximately 4–5 µm), and the pinhole size (4 mm × 0.25 mm). 

Neutron diffraction was used for the determination of the RS profile across the weld in the different depths of the plate (see Figure 1). The measurements were performed in the Institute of Nuclear Physics of Czech Academy of Sciences using the HK4-Strain scanner (ÚJV Řež a.s, Řež, Czech Republic). For this purpose, the wavelength of the radiation used for the diffraction on *{110}* ferrite lattice planes was *λ* = 0.213 nm.

Samples were produced with dimensions corresponding to the ČSN EN ISO 6892-1 standard. Afterwards, tensile tests were performed according to the ČSN EN ISO 4136 standard on the EUS 40 testing machine (Werkstoffprüfmaschinen Leipzig GmbH, Leipzig, Germany). High-cycle fatigue tests were performed on the Schenck PHT resonance machine (Schenck PHT GmbH, Darmstadt, Germany) with a load capacity of 200 kN with Zwick computer-controlled electronics (Zwick Roell Group, Ulm, Germany) and performed at different stress ranges to obtain the whole Wöhler curve, including the endurance limit. The load asymmetry was R = 0.1 and the frequency was 33 Hz. The dog bone-shaped sample was used for fatigue and tensile tests. The weld was located across the center of the length of the analyzed samples where the surface of the sample was as welded. 

A three-point bending impact test was performed according to the ČSN EN ISO 148-1 standard on the PSWO 30 impact hammer (Werkstoffprüfmaschinen Leipzig GmbH, Leipzig, Germany) at a temperature of 0 °C. 

## 3. Results

The standard tests (metallographic and tensile) were supplemented with impact and fatigue tests. These findings were compared with non-destructive X-ray and neutron diffraction, and all gave an extensive and complex description of not only the weld itself but also the surrounding areas.

### 3.1. Metallographic Study

Figure 2 shows the macrostructure of a double-sided welded joint. The weld metal (WM), heat-affected zone (HAZ), and base material (BM) can be distinguished. The width of the WM and the HAZ were 2.2 mm and 1.2 mm, respectively. Apart from the pores, no other defects of the macrostructure, such as cracks, lack of fusion, etc., were observed. Pores arose in the WM due to the instability of the keyhole [18].

The microstructure of the BM consisted of mainly polygonal ferrite and perlite. The ferrite grains were very fine, with a mean grain size of 5.7 µm (G12). No significant changes in ferritic grain size at the surface and or the center of the plate were found. The volume fraction of perlite in the ferrite matrix was estimated at 1%.

The HAZ could be divided into four different zones (see Figure 2b): coarse-grained (CGHAZ), fine-grained (or recrystallized zone—FGHAZ), intercritical (ICHAZ), and sub-critical (or over-tempering zone—SCHAZ) [19]. The closest to the WM was the CGHAZ. This zone was relatively narrow, and the microstructure was heated nearly to the solidification temperatures, resulting in the coarsening of the primary austenitic grains and the decomposition of the carbides and carbonitrides of the microalloying elements, which led to the formation of coarse bainite and small volume fraction of low-carbon martensite (Figure 3a). On the other hand, the FGHAZ was wide in comparison to other zones in the HAZ and consisted of fine ferrite and bainite. The last two zones (ICHAZ and SCHAZ) were characterized by heating from the laser source to temperatures below A_3_, which led to partial recrystallization of the ferrite and transformation of the pearlite.

The WM microstructure was oriented preferentially in the direction of the solidification gradient and consisted of acicular ferrite and grain boundary ferrite (see Figure 3b).

### 3.2. X-ray and Neutron Diffraction

In our previous work [20], the stability of the laser welding process along the entire length of the weld was fulfilled. Therefore, only one area, the middle part of the plate (see Figure 1), was analyzed.

#### 3.2.1. Microstructure Parameters

Resulting from a previous study by Čapek et al. [14], the analysis of the microstructural parameters (e.g. microdeformation *e*, crystallite size *D*, or FWHM parameters) by non-destructive X-ray diffraction in the surface layers indicated that this method could be a good indicator of future microcrack initialization and propagation. 

The comparison of microstructural parameters depending on the distance from the weld axis is depicted in Figure 4. Especially on the bottom side, the typical zones were possible to identify:(1)WM (up to 1 mm from weld axis): The values of *e* and *D* were the highest, approximately 18 × 10^−4^ and above 500 nm (due to the maximum value of the *D* given by the MStruct program [17]), respectively. This zone was very coarse-grained; moreover, each grain was distorted due to the quenching of the weld. Paying attention to the FWHM parameter, it could be stated that the microstructure of the WM was finer on the top side and was more coarse-grained than the CGHAZ.(2)CGHAZ (up to 2 mm from weld axis): The values of *D* were still very high (around 500 nm) and *e* decreased about 30%. The microstructure was still coarse-grained but, due to the smaller heat-load, the grain distortion, i.e., *e*, was smaller.(3)FGHAZ (up to 3.5 mm from weld axis): The steep decreasing of *D* values was observed and the values of *e* started gently decreasing. Parallel with *D* decreasing, the FWHM increased.(4)BM with high *e* (up to 8 mm from weld axis): The values of *D* reached the minimum (bulk) value and *e* was still gently decreasing. In this region, there was a fine-grained microstructure but with relatively high grain distortion. The temperature was not high enough to enlarge the grains; on the other hand, the heat influence and cooling rate were enough to retain sufficient energy for grain distortion.(5)BM with high FWHM (up to 15 mm from weld axis): *e* reached the bulk values. On the other side, the FWHM parameter still had a high value, which indicated the relatively high dislocation density. This zone was still influenced by welding, but the microstructure itself was not.(6)BM (above 15 mm from weld axis): All microstructure parameters reached the constant values. The microstructure was not influenced by welding.

It is necessary to mention that the analyzed widths of the zones were strongly affected by the real irradiated area. Considering the divergence of X-ray radiation and experiment geometry, it was approx. two times larger than the pinhole size. Therefore, very narrow ICHAZ and SCHAZ were not observed and the mentioned zone widths were narrower. The differences between the top and bottom side were caused by double-side welding, where the first weld was annealed during the second pass. 

The boundary between the HAZ and BM created a microstructural notch because of the rapid decrease in *D* values. Another microstructural notch was on the boundary of the WM and HAZ (i.e., fusion zone, FZ). The FZ identification was limited by the high *D* value, so therefore the decreasing of *e* could be appropriate for identification. These areas were the most critical for the potential initialization of surface fatigue crack.

#### 3.2.2. Residual Stresses

From the point of view of residual stress (RS) generation, the thickness of welded plates is a very important parameter because thicker material means more volume of materials for heat conducting (see previous research [20]). Moreover, according to [20], the absolute value of RS also increases with the increasing thickness, which is caused by the mentioned heat conducting. Therefore, in the case of the analyzed sample, the highest thermal influence consequent of the higher shrinkage was closer to the weld in the L direction (see Figure 5).

From the obtained RS profiles, it could be said that the typical trends were generated, i.e., tensile RS in the L direction, and both compressive and tensile RS in the T direction (see Figure 5). Contrary to the expectation, the high tensile RS were found on the top side in the T direction. These stresses were probably generated during the second pass when the deformation occurred (warping). 

The comparison of RS trends in different depths (surface measured by XRD and bulk material by neutron diffraction) showed that longitudinal components have typical tensile character. On the other hand, in the case of transversal components, substantial differences across the thickness were observed. These differences could be explained by the fact that the length of the heated material and the subsequent shrinkage in the T direction was much shorter than in the L direction, and therefore, transformation (compressive) RS could occur in the T direction. These volume changes had a different impact on the surface layers and bulk. Another significant influence was double-side welding and the associated warping in the T direction. Considering these influences, the non-symmetric trends of bulk RS around the weld were observed by neutron diffraction.

The RS trends also showed the size of the so-called stress affected (tempering) zone (SAZ). This zone did not differ from the BM by microstructure but only by RS; it was much wider for the bulk material than for the surface (see FWHM parameter) and extended approx. 25 mm from the weld axis.

### 3.3. Mechanical Tests

#### 3.3.1. Tensile Test

All tested samples reached a satisfying tensile strength of *R_m_* = 556.5 ± 0.5 MPa; thus they showed at least the minimum strength of the BM (490 MPa). Despite the porosity of the weld (see Figure 2), the failure occurred outside the WM. This finding could be explained by the combination of the real effective cross-section and the mechanical properties of the weld, which were (in this case) more dominant than the porosity.

#### 3.3.2. Hardness

Not exceeding the limit values of the hardness was one of the fundamental conditions for approving the welding procedure. Figure 6 depicts the HV5 (Vickers hardness test, load 5 kg) through the laser weld. In general, there are two different standard methods of evaluation, namely, (1) performing indentations at exact regular distances, and (2) a simplified method of performing indentations at each weld zone—the BM, HAZ, WM, second HAZ, and BM again. The latter method was used. According to the standards, three indents were made in each zone. The diagram validates the excellent quality, stability, and reproducibility of the laser welding in the WM and HAZ—the three dependencies, namely in the bottom surface, middle, and top layer, were almost perfectly identical.

The diagram also shows a very uniform and smooth course of hardness from the BM through the HAZ and the WM. When the hardness was recalculated to the strength according to EN standards, the differences of the strength of the weld (around 710 MPa) and base material (550 MPa) were unusually low, which can be considered a very positive factor and significantly smaller than usual for conventional welds. Compared to [6], the difference in hardness between the WM and HAZ was negligible. The reason was in the low heat input during the welding and consequent quenching and the lack of hard martensite formation. It is necessary to note that the increasing hardness could have been caused by a fine-grained microstructure or higher dislocation density.

#### 3.3.3. Impact Test

The average values with the standard deviation of the absorbed energy were high for all tested areas (WM, HAZ, and BM) and exceeded 80 J (see Figure 7). However, for the WM samples, it must be noted that the individual values had a significant scatter; the minimum measured value was 42 J and the highest 125 J. This was caused by a different amount of porosity in the weld and mainly by the deviation of the main crack from the WM to the HAZ and BM. The deviation of the crack from the WM caused the values of the absorbed energy to be artificially increased. This issue was also widely discussed in all the technologies that created narrow weld beads such as laser and electron beam welding. Many of the articles, e.g., [21,22], discussed this problem in detail and it was generally called fracture path deviation (FPD). FPD arose not only in impact tests but also in static three-point bend tests and fracture toughness tests. Different fracture morphology was also observed depending on the FPD (see Figure 8). In the case of the sample with a minimum measured value of 42 J, it can be seen that the fracture had a limited plasticity represented by lateral expansion (see Figure 8a). Dimple morphology also occurred on the samples with higher absorbed energy and cleavage facets on the samples with lower absorbed energy.

#### 3.3.4. High-Cycle Fatigue Test

The results of the high-cycle fatigue (HCF) tests are shown in Figure 9, where the fatigue resistance of the laser welds is compared with conventional arc-welding of the same material and same thickness. The arc welds were investigated in our previous study [10,12]. 

The HCF results of the laser-welded samples were burdened with a very big scatter. However, the analysis of the fracture surfaces enabled the specimens to be divided into two groups. The first one contained significant weld defects—a lack of fusion in the center (see Figure 10a). These defects were areas of quick fatigue crack initiation and growth to failure. The crack initiation in specimens without welding defects occurred on the surface in the FZ (the boundary of the WM and HAZ) (see Figure 10b). Therefore, the crack initiation and growth process were significantly slower, resulting in better fatigue resistance.

## 4. Discussion

From the fatigue life point of view, the critical areas for a potential crack initialization were essential to investigate. Comparing destructive (Figure 2 and Figure 3) and non-destructive (Figure 4 and Figure 5) methods, the typical microstructure zones were observed: the weld metal (WM), the coarse-grained (CGHAZ), fine-grained (FGHAZ), intercritical (ICHAZ), sub-critical (SCHAZ) heat-affected zones, the stress affected zone (SAZ), and the base material (BM). Each zone was recognized according to the microstructure, crystallite size *D*, microdeformation *e*, and residual stresses (RS): WM (coarse-grained, the highest *D* and *e*), CGHAZ (coarse-grained, high *D*, medium *e*), FGHAZ (fine-grained, small *D*, medium *e*), SAZ (fine-grained BM microstructure, small *e*, high FWHM parameter, or non-constant RS), and BM (unaffected by welding). 

Considering the microstructure from Figure 3, it could be stated that there was harder and more brittle acicular ferrite in the WM compared to the softer and tougher bainite in the CGHAZ. 

Regarding the RS, the T direction was more important from the fatigue point of view. In this direction, the main stresses occurred on the longitudinal weld of the pressurized pipe. The higher tensile RS were determined on the surface of the top side, reaching 300 MPa. In combination with the stress amplitude applied during the fatigue test (Figure 9), the real stress could exceed the yield strength and the fatigue life could be reduced. 

Therefore, this boundary between the WM and CGHAZ, the so-called FZ, which contained surface notches, high surface tensile RS (on the top side), and grain-size change, was the most critical area for a potential surface crack initialization. 

Another critical area was located on the boundary of the CGHAZ and FGHAZ, mainly due to the microstructural changes where the coarse-grained bainite and fine-grained ferrite with bainite were in the CGHAZ and FGHAZ, respectively. This area could be weakened by the presence of surface tensile RS on the top side. Contrarily, the presence of compressive RS strengthened this area on the bottom side.

The last and least critical area was located between the HAZ (more precisely the FGHAZ) and the BM (more precisely the SAZ). These zones had different microstructure (fine-grained ferrite-perlite in the BM) and hardness, too, but the toughness was similar within the error. These changes were moderated by the presence of the very narrow transitional zones ICHAZ and SCHAZ. However, a detailed study of the hardness [6] showed the soft area in the SCHAZ. The measurement in this study was not so detailed, but the soft area should be taken into account.

The crack initialization across the weld was not impossible, but the occurrence was not frequent; therefore, the mechanical tests in the L direction were not performed. This type of fracture could be caused by a combination of large external forces and a small effective cross-section, i.e., the presence of weld imperfections, different notches, or very high tensile RS. The analyzed RS near the weld in the L direction (see Figure 5) was higher than the yield strength (approx. 355 MPa for the BM), and approached the tensile strength *R_m_*. This effect was caused by the higher hardness of the weld (see Figure 6), which indicates the occurrence of hard phases with higher yield strength.

From the fatigue life point of view, the imperfections were usually considered to be the most qualitative deficiency of the weld. In case of the lack of fusion or big pores (i.e., smaller effective cross-section), the energy accumulated along these imperfections and afterwards, the internal crack was initialized. For this case, the final rupture could occur across the WM and reduce the fatigue life (see Figure 9). Despite the defects in the weld, the fatigue of the welds with defects was better than the fatigue of the arc-welded samples. The most apparent reasons were a huge heat input in combination with microstructural and surface notches. 

Detailed analyses showed that in all cases without defects, cracks were initiated in the FZ on the top side (see Figure 10b). The previous text explained this phenomenon. High surface tensile RS in the transversal direction in combination with surface and microstructure notches, and changes in hardness, toughness, and microstructural parameters *D*, *e*, and FWHM, led to the weakening this zone against fatigue. Only the presence of the tensile character of RS was unusual. These undesirable stresses were probably generated during the second pass when the deformation occurred (warping). This effect indicates that the role of RS on fatigue crack initiation can be considered significant.

## 5. Conclusions

The purpose of this investigation was to explain the relationship of the microstructure, mechanical properties, and residual stresses of laser-welded pressure vessel steel plates. Specifically, 10 mm thick double-sided square butt welds performed on P355NL1 steel plates for high-pressure vessels were investigated. Various experimental techniques were used for the analysis of the weld properties and the critical areas for a potential surface crack initialization. 

The major experimentally obtained knowledge can be summarized in the following points:(1)Despite the internal pores, the failure occurred outside the weld metal (WM) during the tensile test.(2)A uniform and smooth course of hardness from the base material through the heat-affected zone (HAZ) and the WM was found.(3)Using metallography and X-ray diffraction, the microstructure notches were found near the weld. The main notch, the fusion zone (FZ), was located in the boundary of the WM and HAZ.(4)The state of residual stresses (RS) was satisfactory in the bulk and on the bottom surface of the welded plate in the T direction. The unusual high tensile RS were found on the top side in the T direction.(5)The toughness of the WM was smaller due to the presence of hard bainite.(6)The lack of fusion, which was found on the fracture surface, was the primary reason for the fatigue crack in the WM. Nevertheless, the fatigue of the welds with defects was better than the fatigue of the arc-welded samples.(7)The fatigue cracks of the samples without a lack of fusion were always initialized on the top side in the FZ.

The FZ was marked as the most critical area for the potential surface crack initialization because of the higher hardness, different microstructure, presence of notches, various toughness, and unusually high surface tensile RS. All fatigue cracks were initialized on this marked area. Some of the notches could be removed by a smaller heat input during welding. In the case of the RS, the warping of the weld was necessary to prevent. 

Based on these and our previous results [10], laser welding is comparable to electron welding with less manufacturing limitations and, furthermore, has many advantages against conventional metal arc welding. Therefore, the laser is a suitable and promising tool for welding pipelines.

## Figures and Tables

**Figure 1 materials-14-00131-f001:**
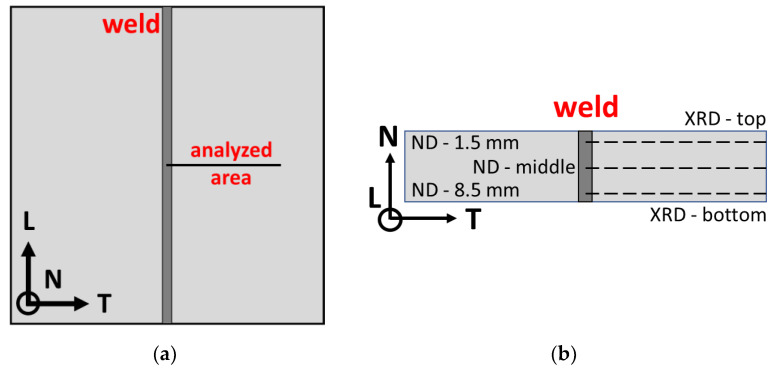
(**a**) Scheme of the sample and (**b**) scheme of marking the depths at which the residual stresses were determined by X-ray (XRD) and neutron (ND) diffraction.

**Figure 2 materials-14-00131-f002:**
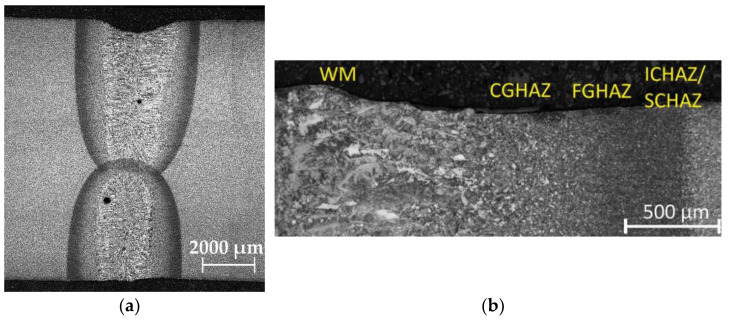
Macrostructure of (**a**) the weld cross-section and (**b**) particular zones after etching.

**Figure 3 materials-14-00131-f003:**
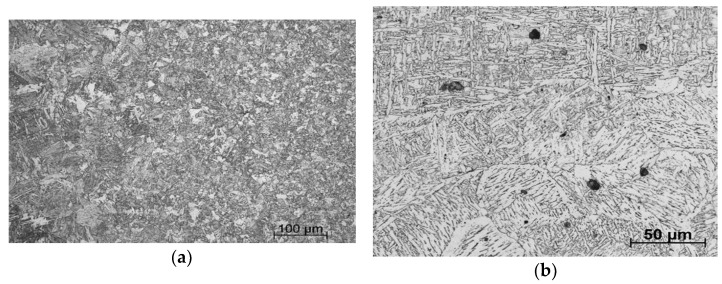
Microstructure in (**a**) the heat-affected zone and (**b**) the weld metal.

**Figure 4 materials-14-00131-f004:**
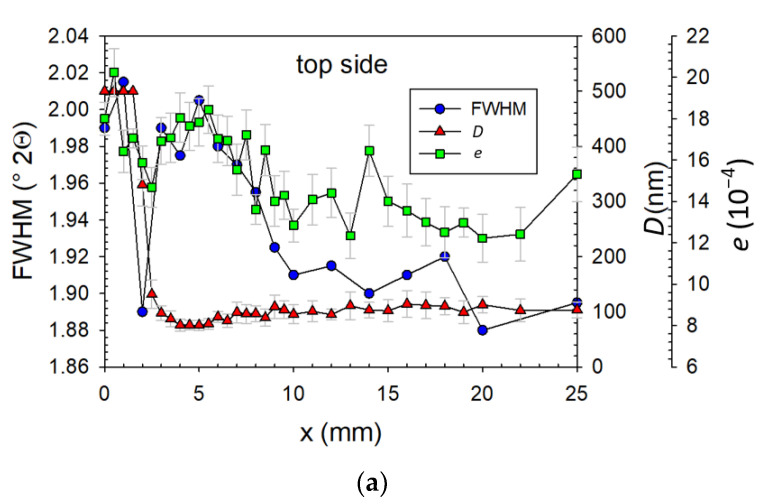
(**a**,**b**) Dependence of FWHM parameter, crystallite size *D*, and microdeformation *e* on the distance from the weld axis.

**Figure 5 materials-14-00131-f005:**
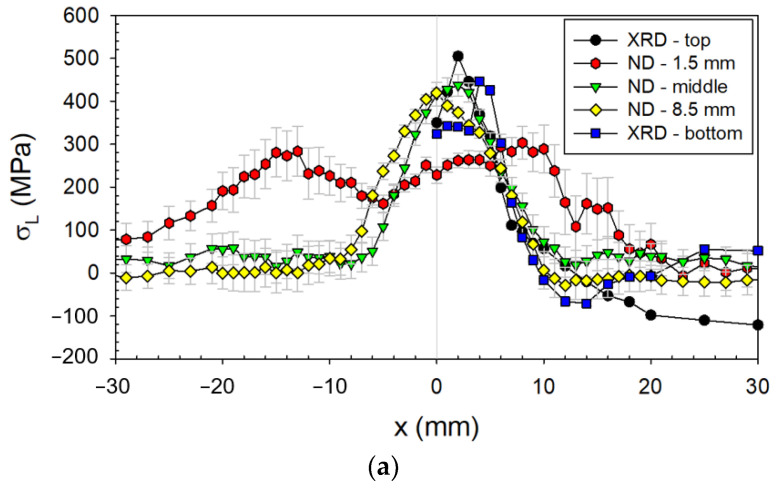
(**a**,**b**) Residual stresses RS (analyzed by XRD and ND) depending on the distance from the weld axis.

**Figure 6 materials-14-00131-f006:**
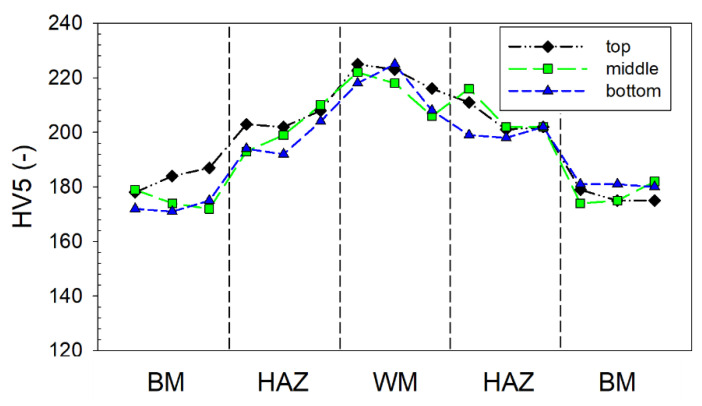
Course of the hardness of the HV through laser welding in the specimen.

**Figure 7 materials-14-00131-f007:**
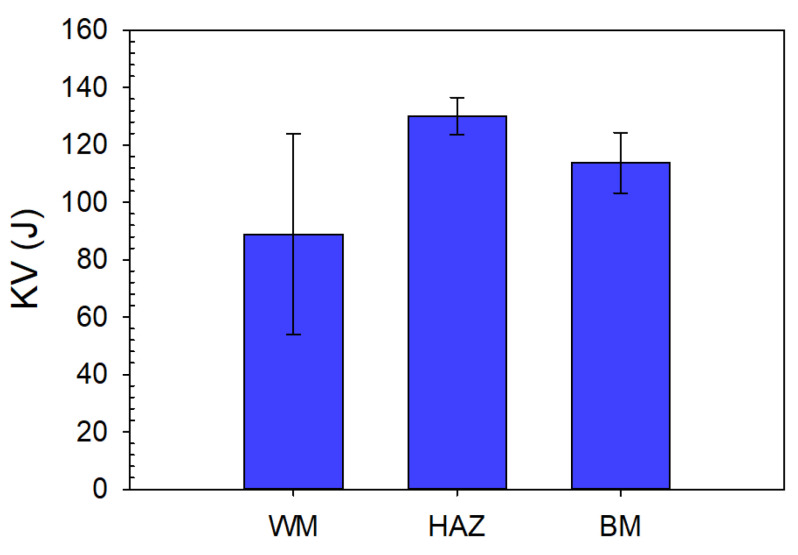
Absorbed energy of the WM, HAZ, and BM during the impact test.

**Figure 8 materials-14-00131-f008:**
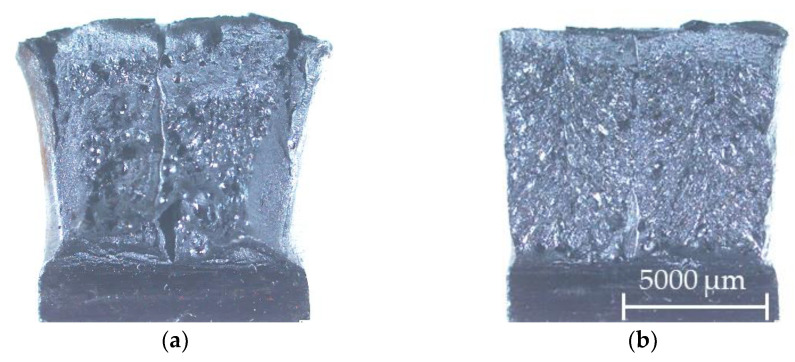
Differences in the fracture morphology of the WM samples: (**a**) 42 J and (**b**) 125 J.

**Figure 9 materials-14-00131-f009:**
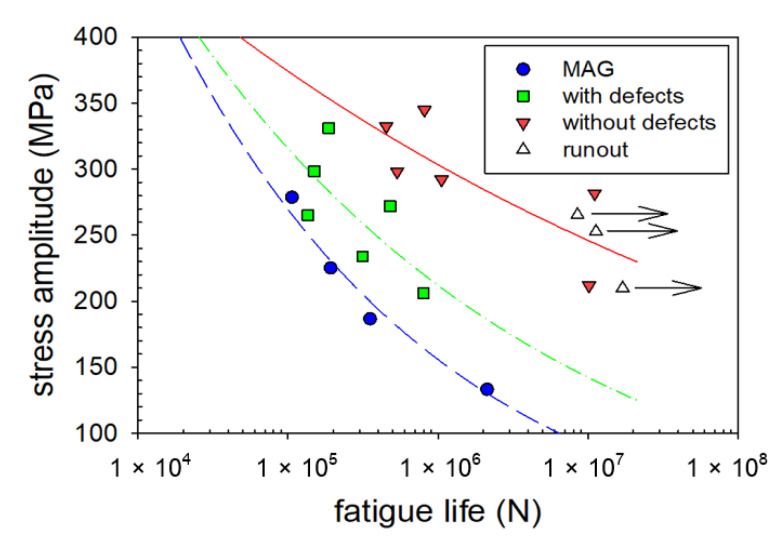
Wöhler curves of laser welded samples with and without internal defects compared with manual arc welding (MAG).

**Figure 10 materials-14-00131-f010:**
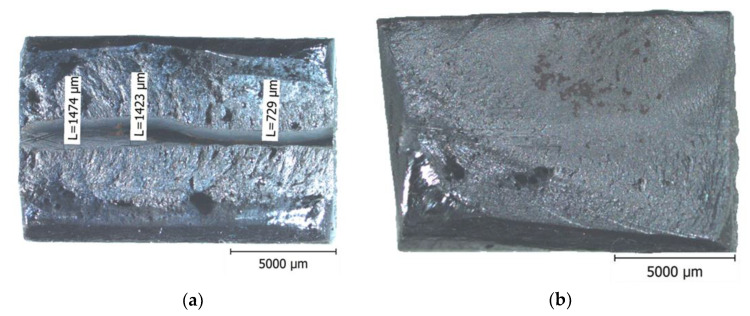
(**a**) Fatigue crack initiation on the weld defect in the center and (**b**) the fracture surface without weld defects—initiation on the top side of the FZ.

**Table 1 materials-14-00131-t001:** Chemical composition of the P355NL1 steel, determined using glow-discharge optical emission spectroscopy.

Element	Fe	C	Mn	Si	Ni	Cr	Cu	V	Al
Weight fraction (wt.%)	98.92	0.067	0.8	0.004	0.005	0.021	0.018	0.002	0.037

**Table 2 materials-14-00131-t002:** Welding parameters.

*P* (W)	*v* (mm∙s^−1^)	Mode	*λ* (nm)
3000	5.5	Continuous	900–1080

## Data Availability

Data sharing is not applicable to this article.

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
