# Peer review of "On the Weldability of Thick P355NL1 Pressure Vessel Steel Plates Using Laser Welding"

_materials, 2020, doi:10.3390/ma14010131_

Round 1

Reviewer 1 Report

The paper is dealing with the laser welding performance of thick plates. The paper is well written and nicely presented. The manuscript could be suitable for publication in Materials after some minor changes.

1. It should clarify why the authors choose to investigate plates while the abstract and introduction are concentrate to explain the requirement of thick pipes as pressure vessels.
2. In section 2 raw 108 “The double-sided square butt weld (the top side was welded first) …”, certainly it could fit a plate that could weld in both side but could not be applicable to long pipes that can be weld in the outer side only. In this manner the electron beam mention in line 45 could be an option as well. Please explain
3. The authors use the term weld metal (WM) for the melt pool, the common term for it in the welding community is Fusion Zone (FZ), please update
4. The use of 3000W Continues laser is generally used to weld no more than 4.5 mm thick while for overlap of the weld is require a depth of more than 5 mm.
5. Figure 2a there is a gap between the double side weld !!! only the HAZ cross each other! The authors should enlarge the matching area and explain, that’s probable cause the poor fatigue result
6. Following my privies comments rows 383 and 384 is irrelevant – there is no comparison in the paper between Laser welding, electron welding, and conventional metal arc welding. This sentence should be removed.
7. The conclusion for welding e pipelines in this paper is meaningless since the weld part testes were plate and weld in both side, in order to conclude and compare to thick pipes the authors require to weld in one side only and for 10 mm thick should use at least 6KW laser
8. Last but not least, the paper could have matched more impact with a relevant simulation of the current laser welding process.

Reviewer 2 Report

Dear sir or madam,

thank you very much for a very comprehensive study on double-sided laser welds for steels used in pipeline-equipment.

As the double-sided welding approach induced an annealing of the material welded in the pass beforehand, would it be interesting to use a laser with higher power to achieve a single-pass weld with full penetration?

Other than that, I would only recommend to spell-check again.

Reviewer 3 Report

This work is very detailed and written well suitable for publication. The combined structure of the grain morphology and residual stress concentration in the weld center shows the impact on hardness, fracture toughness and cracking initiation. The crack without weld-induced defects has higher fatigue resistance. Two recommendations: 1) grain size (Hall-Petch effect) and dislocation density could affect the hardness across the regions; 2) Adjustment in the laser spot size and working distance could significantly change the solidification in the fusion zone and affect grain recrystallization. Is there a good reason to select the current laser welding parameters or any better setups could be made to enhance the weldability? 

Reviewer 4 Report

Dear Authors,

Your manuscript deals with a practical and current issue: the weldability of steel intended for the construction of pressure vessels in the case of double-sided laser welding. The manuscript made a very good impression on me, it contains a lot of relevant information. In my opinion, the work should be published after making the following changes:

I suggest adding the name of the steel grade to the title.

Please add quantitative results to your abstract.

Introduction is well written. I propose to add information on the possibility of making dissimilar joints of high quality with laser welding. It is currently one of the important research trends in welding. I can recommend the following works (not mine) in this topic: https://doi.org/10.3390/ma13204540, https://doi.org/10.1007/s00170-020-04935-5

Line 59: "weld metal under base metal": this is an unclear phrase for me.

Line 64: it should be added that this applies to welding of low-alloy steels.

Line 78: please name the process: "SMAW"

Chapter 2:

Line 103: correct: mm

Line 107: please add that this has a consequence in lowering the value of the carbon equivalent. Please add the Ce value according to IIW in Table 1.

What are the strength properties of the tested steel according to certificate? This information is needed as a criterion to evaluate the results of the presented research.

How many samples were there? How were they taken (scheme?)

Add spaces before units.

According to the journal’s guidelines, the names of the manufacturers and their addresses should be provided for all used devices.

Have VT been performed?

Line 154: change: "weld" to "welded joint".

Line 157: what does “cold joints” mean? I am convinced that the name does not appear in the standards.

Figure 6: "root weld" for a double-sided joint does not sound good.

What is the size of the error in the bars in Figure 7? Standard deviation?

Line 290: Where did the results for MAG welding come from?

This entry is not correct: "[20,21, etc.]"

Conclusion 6: it should be: "lack of fusion".

Supplement your literature sources with the latest articles.

Round 2

Reviewer 4 Report

Dea Authors, 

Thank you for your response and taking into account my most important comments.